# Temperature dependence of mosquitoes: Comparing mechanistic and machine learning approaches

**Tejas S. Athni**[1,2]*, **Marissa L. Childs**[3,4], **Caroline K. Glidden**[2,5], **Erin A. Mordecai**[2]

**1** Harvard Medical School, Boston, Massachusetts, United States of America, **2** Department of Biology, Stanford University, Stanford, California, United States of America, **3** Emmett Interdisciplinary Program in Environment and Resources, Stanford University, Stanford, California, United States of America, **4** Center for the Environment, Harvard University, Cambridge, Massachusetts, United States of America, **5** Stanford Institute for Human-centered Artificial Intelligence, Stanford University, Stanford, California, United States of America

* tathni@hms.harvard.edu

**Data Availability Statement:** Data and code can be found at https://github.com/tathni/mosquito-sdm.

**Funding:** TSA is supported by the National Institute of General Medical Sciences (grant no.

## Abstract

Mosquito vectors of pathogens (e.g., *Aedes*, *Anopheles*, and *Culex* spp. which transmit dengue, Zika, chikungunya, West Nile, malaria, and others) are of increasing concern for global public health. These vectors are geographically shifting under climate and other anthropogenic changes. As small-bodied ectotherms, mosquitoes are strongly affected by temperature, which causes unimodal responses in mosquito life history traits (e.g., biting rate, adult mortality rate, mosquito development rate, and probability of egg-to-adult survival) that exhibit upper and lower thermal limits and intermediate thermal optima in laboratory studies. However, it remains unknown how mosquito thermal responses measured in laboratory experiments relate to the realized thermal responses of mosquitoes in the field. To address this gap, we leverage thousands of global mosquito occurrences and geospatial satellite data at high spatial resolution to construct machine-learning based species distribution models, from which vector thermal responses are estimated. We apply methods to restrict models to the relevant mosquito activity season and to conduct ecologically plausible spatial background sampling centered around ecoregions for comparison to mosquito occurrence records. We found that thermal minima estimated from laboratory studies were highly correlated with those from the species distributions (r = 0.87). The thermal optima were less strongly correlated (r = 0.69). For most species, we did not detect thermal maxima from their observed distributions so were unable to compare to laboratory-based estimates. The results suggest that laboratory studies have the potential to be highly transportable to predicting lower thermal limits and thermal optima of mosquitoes in the field. At the same time, lab-based models likely capture physiological limits on mosquito persistence at high temperatures that are not apparent from field-based observational studies but may critically determine mosquito responses to climate warming. Our results indicate that lab-based and field-based studies are highly complementary; performing the analyses in concert can help to more comprehensively understand vector response to climate change.

T32GM144273). MLC was supported by the through the Illich-Sadowsky Interdisciplinary Graduate Fellowship program at Stanford University and an Environmental Fellowship at the Harvard University Center for the Environment. EAM and CKG were supported by the National Science Foundation and the Fogarty International Center (grant no. DEB-2011147). EAM was additionally supported by the National Institute of Allergy and Infectious Diseases (grant nos R01AI168097 and R01AI102918), the National Institutes of Health (grant no. R35GM133439), and by seed grants from the Stanford Woods Institute for the Environment, King Center on Global Development, Center for Innovation in Global Health and Terman Award. CKG was additionally supported by a Stanford Institute for Human-centered Artificial Intelligence Postdoctoral Fellowship. The funders did not play a role in study design, data collection and analysis, decision to publish, or preparation of the manuscript. Funder websites in include: https://www.nigms.nih.gov/ https://vpge.stanford.edu/fellowships-funding/sigf/ sigf-named-fellowships https://environment. harvard.edu/environmental-fellows-program https://www.nih.gov/about-nih/what-we-do/nih-almanac/fogarty-international-center-fic https:// www.niaid.nih.gov/ https://www.nih.gov/ https:// kingcenter.stanford.edu/ https://globalhealth. stanford.edu/ https://biology.stanford.edu/news/ erin-mordecai-receives-terman-award https://hai. stanford.edu/research/fellowship-programs https:// woods.stanford.edu/.

**Competing interests:** The authors have declared that no competing interests exist.

## Author summary

Mosquito vectors are strongly affected by temperature, and their distributions are likely to shift under climate change. Lab studies show that mosquito abundance has a unimodal response to temperature with thermal optima, upper and lower thermal limits. However, it remains unknown how mosquito laboratory-derived thermal responses relate to the thermal responses of mosquitoes in nature. We used a global database of field-collected mosquito occurrences, geospatial environmental covariates, and species distribution models to estimate the relationship between temperature and probability of mosquito occurrence. We found that thermal minima (r = 0.87) and, to a lesser degree, thermal optima (r = 0.69) estimated from laboratory studies were correlated with those from the species distribution models. For most species, we did not detect thermal maxima. These results suggest that laboratory studies and field-based machine learning studies are complementary. Together, they can help to better understand vector response to climate change.

## Introduction

Mosquito-borne diseases (e.g., malaria, dengue, Zika, chikungunya, West Nile, yellow fever) are responsible for a significant worldwide burden of infectious disease and represent a major threat to global public health [1–5]. As small-bodied ectotherms, mosquito vectors are sensitive to environmental conditions and, in particular, to abiotic factors such as temperature [6,7]. Climate change is likely to alter the climatic and habitat suitability for and the geographic distribution of mosquitoes, in turn affecting the distribution of pathogens they transmit [8–11]. Understanding the limitations that temperature and other ecological conditions place on mosquito vectors is critically important for predicting how mosquitoes and vector-borne diseases will respond to climate change.

Previous laboratory experiments have measured the relationship between temperature and mosquito life history traits such as biting rate, adult mortality rate, mosquito development rate, and probability of egg-to-adult survival for many vector species [12–15]. Together, these trait thermal performance relationships can provide a mechanistic estimate of equilibrium mosquito abundance as a function of temperature [6,10,16]. For mosquitoes for which these relationships have been estimated, the thermal response curves of individual traits follow predictable patterns: they decline to zero at lower thermal minima and upper thermal maxima and peak at intermediate thermal optima, as expected from first principles of physiology and enzyme kinetics [16–18]. In aggregate, modeled population-level mosquito abundance has a similar unimodal relationship with temperature [6], reflecting the underlying life history traits [10]. These studies confirm a core tenet of the metabolic theory of ecology, which states that organismal physiology operates within and is restricted by thermal limits [19]. Temperature-dependent effects of these life history traits in turn affect transmission of mosquito-borne disease [6,20–33]. Despite these clear predictions from ectotherm physiology and laboratory thermal performance experiments, it remains unknown to what extent these experimental, laboratory-based mosquito thermal responses predict the realized thermal responses of mosquito populations in the field. In particular: (1) is temperature an important predictor of mosquito distributions? If so, (2) are effects of temperature on mosquito occurrence nonlinear? And if so, (3) do the predicted thermal optima and limits from laboratory experiments align with thermal optima and limits measured in the field? Altogether, our study aims to identify how the gold-standard of measuring mosquito thermal tolerance (lab-based thermal

performance curves) and advancing artificial intelligence efforts can be combined to identify how mosquitos will respond to global change within their natural environment.

The rapid rise of biodiversity data, machine learning, and open source satellite imagery in the past decade has enabled new types of approaches necessary to address these questions. Species distribution models offer a way to connect data on species occurrences with environmental covariates to understand how environmental conditions influence the probability of a species occurring in a given location (i.e., habitat suitability). Due to the challenges of accurately detecting a species' absence, species distribution modeling typically compares locations in which a species has previously been identified (occurrences) and an artificially-generated set of background points (pseudo-absences) which approximate the potentially-habitable and accessible area for a given species [34–37]. Species distribution models have been used to characterize the ecological niches (i.e., habitat and climatic requirements) of organisms across the tree of life, including elephants in South Asia [38], lynxes in Canada [39], ants in Australia [40], and rare plants in California [41]. Beyond predicting geographic ranges, these models can indicate which environmental covariates are most important for predicting species occurrences, and the functional relationships between environmental conditions and species distributions.

Species distribution models often rely on remotely sensed satellite imagery to provide information about biotic, abiotic, and anthropogenic variables. Recent advances in storage and processing of remotely-sensed imagery, including publicly-available platforms like Google Earth Engine [42], have increased the resolution and spatial extent over which these models can feasibly be run. In parallel, the Global Biodiversity Information Facility (GBIF) leverages crowdsourced data collection and digitized biodiversity data, where occurrence points of many species are compiled within a central repository [42]. Moreover, new algorithms and advances in machine learning have improved the predictive capacity of species distribution models, and have facilitated the development of more complex models with interactive and nonlinear relationships between predictors and responses that can potentially capture ecological complexity. Together, these innovations provide ripe new avenues and abundant data through which to tackle ecological questions.

Despite the promising advances in species distribution modeling, major limitations also remain. First and most importantly, species distribution models are correlative and not causal, so the factors that best capture distributional limits (i.e., discriminate between presences and pseudo-absence) may not fully capture the true physiological and ecological limits. This may occur because biologically limiting factors are not easily measured or highly variable across distributional ranges, because may have high-order and non-linear correlations with other factors, or because organisms have not yet encountered their limits in a given ecological factor due to constraints posed by other factors. For example, a temperature limit may not be easily identifiable from species distribution models because a species is constrained to a habitat type where such limiting temperatures do not currently occur. Alternatively, species that are actively expanding their range may not have yet expanded to occupy the entire available suitable habitat. This means that species distribution models have limited capacity to extrapolate future changes in ecological constraints on a changing world, without complementary knowledge from mechanistic approaches such as experiments and mechanistic models. In this work, we build on the complementary strengths of two distinct approaches—the biological mechanism and interpretability of simple mathematical models based on controlled laboratory experiments, and the complexity and flexibility of species distribution models that capture interactive and nonlinear environmental relationships—to ask whether temperature, a fundamental biological constraint, has similar inferred effects on the distributional limits of key mosquito vector species.

We focus on the vectors of the world's highest-burden mosquito-borne diseases (malaria, dengue, chikungunya, Zika, West Nile, and other arboviruses)—*Aedes aegypti*, *Aedes albopictus*, *Anopheles gambiae*, *Anopheles stephensi*, *Culex pipiens*, *Culex quinquefasciatus*, and *Culex tarsalis*—each of which has well-characterized thermal performances curves in laboratory settings and a wealth of data on occurrences in the field [6,9,16,25,28,43,44]. *Ae. aegypti* and *Ae. albopictus* are important vectors of dengue, chikungunya, yellow fever, and Zika viruses. *An. gambiae* and *An. stephensi* primarily vector the *Plasmodium* spp. protozoans that cause malaria. *Cx. pipiens*, *Cx. quinquefasciatus*, and *Cx. tarsalis* transmit West Nile, St. Louis encephalitis, Western Equine encephalitis, and other zoonotic viruses, and *Cx. quinquefasciatus* additionally transmits lymphatic filariasis.

While previous mosquito species distribution models have focused on estimating habitat suitability and predicting occurrence [8,9,44], our goal is different: to specifically dissect the relationship between temperature and probability of mosquito occurrence, while incorporating other constraints on occurrence, and to compare this relationship among mosquito species and with laboratory-based model predictions. For this reason we created new models, rather than using published ones, that use consistent methods across multiple mosquito species. Further, we aimed to construct species distribution models for each focal species using spatially- and temporally-similar data sources and consistent model assumptions in order to compare thermal dependence across species.

We make two important methodological advances to ensure that our species distribution models capture true thermal limits on mosquito occurrence, rather than sampling bias or restricted geographic ranges that covary with temperature. First, we sample background pseudo-absence points only from the set of ecoregions in which the focal mosquitoes occurred, as well as adjacent ecoregions. This ensures that comparator pseudo-absence points are selected from regions where the focal mosquito species could have realistically been found but were not, or in other words, zones that are ecologically plausible for an occurrence. Second, we restrict temperature measurements to an 'activity season' during which each mosquito species is blood-feeding and reproducing, and not in dormancy or torpor. This captures temperature constraints within a physiologically realistic period, rather than constraints due to overwintering or drought persistence, which are less likely to be comparable to laboratory-based trait thermal response experiments. By using global mosquito occurrences from 2000–2019, geospatial satellite-derived covariates at high spatial and temporal resolution, bias-reducing methods described above, and a gradient-boosted classification tree machine learning algorithm well-suited for prediction tasks, we aim to provide evidence that spans across continents and decades for the globally important vectors of human malaria and arboviral disease. By leveraging both approaches, we may be able to use their complementary strengths to minimize each of their limitations. With a greater understanding of thermal constraints on vector species occurrence, we can improve the ability to accurately project and mitigate the future impacts of climate change on mosquito-borne disease distributions.

## Methods

### Overview and study period

We focused on 2000–2019 to isolate recent land-use and climate patterns but provide a long enough time period to capture mosquitoes' stable spatial distribution. The environmental covariates and mosquito occurrences were both extracted for this time period, providing temporal consistency in the analysis [45]. An overview of the methods can be seen below (Fig 1). Computation was performed in R v4.1.1, Google Earth Engine and the Sherlock computing cluster at Stanford University.

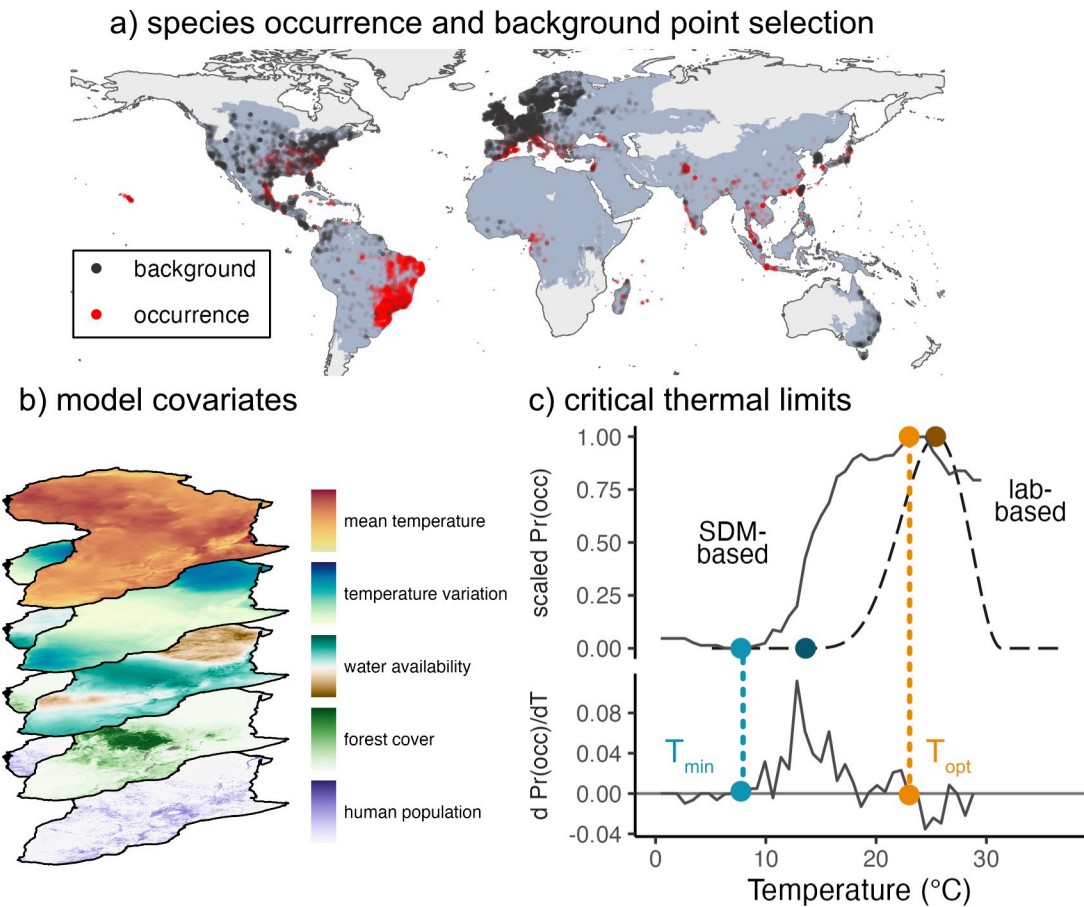

**Fig 1. Methods for comparing mosquito thermal performance derived from species distribution models (a-c) and laboratory trait-based models (c).** The analysis involved two major steps: statistical modeling of mosquito occurrence using global species occurrences (a) and geospatial data (b) to identify the relationship between temperature and species occurrence (solid line, c), and comparison with mosquito abundance curves previously derived from laboratory life history traits [16] (dashed line, c). Temperature-dependent mosquito abundance (M(T); dashed line in c) is modeled as a function of temperature-dependent eggs laid per female mosquito per day (EFD(T)), probability of egg-to-adult survival (pEA(T)), development rate from egg to adult (MDR(T) = 1/development time in days), and daily adult mosquito mortality rate ($\mu$(T)). Trait curves adapted from [16,31]; M(T) equation originally derived from [6]. Thermal minima and optima of the probability of occurrence from the species distribution models were identified by calculating the temperature at which the empirical derivative was first positive and stayed positive for the next step (light blue point, c) and the temperature where the empirical derivative was zero and the probability of occurrence was at its maximum (light orange, c). Sources of data for the stack of covariates are described in Table B in S1 Text. The background shapefiles are based on from https://ecoregions.appspot.com/ and coastlines are from https://ec.europa.eu/eurostat/web/gisco.

## Environmental predictors

Environmental covariates were defined *a priori* in order to avoid data-dredging [11], and were based on a literature search of previous species distribution models. Variables were informed by ecological and biological relevance, expectation to play a role in mosquito vector ecology, and top contributor status in previous mosquito species distribution models [46–49]. Environmental covariates were computed using remotely sensed and reanalysis data, and resampled using bilinear interpolation to consistent 1 km x 1 km resolution on Google Earth Engine. This approach using global satellite data avoids the spatial interpolation limitations of common climatic datasets such as WorldClim, in which data are sourced from ground meteorological stations with uneven geographic distribution and nonrepresentative selection.

**Table 1. Mosquito species by activity season.**

| Mosquito Species | Activity Season (Reference) | Raw Number of Occurrences | Unique Cell Centroids |
| --- | --- | --- | --- |
| *Aedes aegypti* | Year Round [57,58] | 37,115 | 9,299 |
| *Aedes albopictus* [#] | Photoperiod [52,59] | 39,488 | 8,688 |
| *Anopheles gambiae* | Precipitation [51,55] | 13,650 | 903 |
| *Anopheles stephensi* | Year Round [60] | 1,232 | 358 |
| *Culex pipiens* | Photoperiod [53,54] | 98,276 | 1,949 |
| *Culex quinquefasciatus* | Year Round [61,62] | 30,978 | 2,670 |
| *Culex tarsalis* | Photoperiod [63] | 44,496 | 909 |

[#] This species enters winter diapause in some regions such as the Northeast US, but not other regions like the Southeast US.

In order to verify minimal overlap between predictor variables, a pairwise correlation analysis was performed with a Pearson's correlation coefficient r < |0.8| threshold (Fig A in S1 Text). Annual averages for covariates were computed across the study period, providing the characteristic climatic conditions across the study's temporal range.

**Temperature predictors.** While many species are active year-round (Table 1), numerous species physiologically undergo photoperiodic diapause in the winter months when ambient light is low in order to conserve energy for life sustenance [50] or aestivate during the dry season when no breeding habitat is available [51]. To capture the temperature that mosquitoes experience while active, temperature covariates were calculated over the activity season.

For the purposes of this study, the photoperiodic activity season was defined as the time period encompassing days with at least 9 hours of sunlight (Fig B in S1 Text). This definition of the activity season was chosen based on a variable separate from temperature and on laboratory studies that found that either 8 or 9 hours of light can induce diapause in mosquitoes in the absence of temperature change [52–54]. The photoperiod-based constraint only limited the activity season in higher latitudes and did not affect lower latitudes (Fig B in S1 Text). Other tropical mosquito species (e.g., *Anopheles gambiae*) do not enter photoperiodic dormancy due to their equatorial setting. Rather, these species' activity seasons are instead constrained by lack of precipitation, when insufficient habitat is available for immature development and survival. Studies have highlighted how Sahelian *Anopheles gambiae* mosquitoes virtually disappear in the dry season then re-establish in the wet season through aestivation behavior [51,55], and climatological studies in the Democratic Republic of the Congo characterize the dry season as having less than 50 mm of rain in the past month [56]. Precipitation activity season is therefore defined in our study as the time period in which there is, on average, at least 50 mm of precipitation in the past 30 days (Fig C in S1 Text).

Predictors capturing thermal central tendency include year round temperature mean, photoperiod activity season temperature mean, and precipitation activity season temperature mean. Measures of seasonal thermal variation include year round temperature standard deviation, photoperiod activity season temperature standard deviation, and precipitation activity season temperature standard deviation.

**Other environmental predictors.** Environmental variables other than temperature included measures of vegetation cover, such as enhanced vegetation index mean, enhanced vegetation index standard deviation, and forest cover percentage, to capture land cover that might affect mosquito habitat suitability (Table B, Fig D in S1 Text). To determine which environmental predictors to include, we first looked to the literature to understand which ecological variables were consistently utilized in past SDM models and which were found to be the most important predictors in other papers for our target mosquito species. Among those top

predictors from previous papers, we selected covariates that could best capture the underlying vector ecology and biology.

For example, in this process, we identified that human population density is a variable that may be ecologically important as it represents the fact that many mosquito species are highly human-dependent with human blood-feeds, inhabit urban or peri-urban niches, and breed in artificial containers or pools of water in close proximity to human settlements. Indeed, human population density was the topmost predictor for in models for *Ae. aegypti* [48], and for *Cx. pipiens* [47]. Precipitation of the driest quarter, on the other hand, was thought to capture suitable mosquito breeding habitats in the setting of less standing water in the dry season, and has been found to be an important predictor for *An. gambiae* [46], for *Ae. aegypti* and *Cx. pipiens* [49]. Enhanced vegetation index describes the quality of vegetation features like leaf area, canopy cover, and sugar resources that may provide alternate food sources or resting sites for many species [47], and both mean and standard deviation of vegetation index have important predictors for *Cx. pipiens* in a previous model [47].

Ultimately, human population density and cattle density were included to capture sources of blood meal, anthropophilic hotspots, and human-seeking behavior. Precipitation of the driest quarter, precipitation of the wettest quarter, and surface water seasonality were included to capture hydrology-driven characteristics influencing mosquito breeding site availability or abundance, while wind speed was included to capture potential aerial dispersal limitation of mosquitoes and relative humidity was included because it affects survival and life history [64]. Precipitation of the driest quarter and precipitation of the wettest quarter were defined as the set of 3 consecutive months with the least and most precipitation, respectively, across the 20-year study period.

## Occurrence and background

**Presence occurrences.**   Focal mosquito species were selected based on global disease burden, availability of life history trait data from published laboratory studies, and abundance of publicly-accessible species occurrences, resulting in seven species: *Ae. aegypti*, *Ae. albopictus*, *An. gambiae*, *An. stephensi*, *Cx. pipiens*, *Cx. quinquefasciatus*, and *Cx. tarsalis*. Collectively, these seven species span nearly all habitable continents (Figs E-K in S1 Text). Species occurrence records for each species were obtained from the Global Biodiversity Information Facility (GBIF) for the years 2000–2019. Supplemental occurrence points were obtained from two papers for *An. stephensi* [65] and *An. gambiae* [66], which had low sample sizes from the GBIF database. Occurrences explicitly tagged in Africa were discarded from the supplemental *An. stephensi* data, as these are part of an ongoing species invasion outside of the native range of the species in South Asia [65], and species distributions models have limitations in detecting environmental suitability for invasive species as they do not yet occupy their environmental niche and are not yet in equilibrium with their environment [67].

Raw datasets were cleaned to remove occurrences with unknown basis of record and those obtained through fossil records, due to the temporal and geographic uncertainty between when and where the organism actually existed during its lifetime and its location of fossilization. Uncertainty in the latitude and longitude coordinates of occurrence points were taken into account through a two-step process. First, points with a coordinate uncertainty less than 1000 meters—the diameter of our environmental covariate grid cell size—and those points with missing coordinate uncertainty were selected. Of the points with missing coordinate uncertainty, we retain only those points with a latitude and a longitude with at least two significant decimal points (i.e., each hundredth corresponds to approximately 1.11 km). The occurrence data were then restricted to points that reside on landmasses and not in the ocean, those

with non-missing values for all covariates, and, when relevant, and those that occurred in areas where the species' activity season length was greater than 0 (i.e., when the precipitation was above a certain value in the grid cell, i.e., precipitation activity season, or when there was a certain amount of daylight hours in the grid cell, i.e., photoperiod activity season) (Figs B-C in S1 Text). We filtered the occurrences to the cell centroids of the unique 1 km x 1 km cells in which occurrence points were obtained, which serves to spatially thin the occurrence points and reduce the effect of biased sampling [68] (Table 1; Table A in S1 Text).

**Pseudo-absence background sampling.**   Species distribution models work by comparing the environmental covariates that best predict locations with species occurrences to those in which the species does not occur, to understand the environmental conditions that constrain a species' distribution. As the majority of information is on species occurrences rather than species absences, methods for species distribution modeling have developed to sample pseudo-absences: points that represent the area in which a species could have been sampled yet was not reported. In order for these sampled points to serve as plausible points where the species of interest was not found, they should be within the species' accessible area, or the region that is reasonably reachable by the species through dispersion or migration over the relevant time period [34,36,37]. Choosing pseudo-absences that occur within the same ecoregions, i.e., geographic regions that possess similar species and community assemblages, as occurrences provides a way to geographically restrict possible background points. Moreover, these points should be chosen in a way that takes into account sampling bias and collection effort.

To sample background points, we first created a bias mask to probability-weight the sampling. 72 million occurrences from Class Insecta, filtered to the same criteria as mosquito occurrence points (i.e., <1000m coordinate uncertainty or at least two significant decimal points of latitude and longitude, 2000–2019 study period, no fossil record or literature points), were extracted from GBIF. In order to capture the frequency of GBIF sampling conducted in a specific geographic location, the number of insects were tallied for each 1 km x 1 km grid cell, and these counts were converted into proportions of all insects. This proportion was regarded as the probability weight for that given grid cell. To provide adequate sample size while avoiding a skew of the overall sample towards disproportionately favoring pseudo-absences over occurrences, we selected a number of unique background cell centroids equal to twice the number of unique occurrence centroids [69]. Background cells were sampled at random without replacement and weighted by the bias mask from the geographical region that was bounded by the focal species' ecoregions of occurrence. In addition, we include adjacent ecoregions in the background sampling space to promote contiguity in the potential ecological spaces from which pseudo-absences can be drawn and to better encapsulate the broad range of environmental covariates that a given species may experience in nature. Ecoregions were defined as RESOLVE Ecoregions that intersected with a given species' occurrence centroids, buffered with a 200 km radius buffer that approximated an upper end of the wind-assisted geographic mobility range of a mosquito [70,71]. By buffering occurrences and then choosing the set of ecoregions and adjacent ecoregions in which these buffered points lie, we approximate what is known in ecological theory as the 'accessible area,' described above [34,36,37]. Because our focal questions centered on species' thermal breadth, we wanted to ensure that as broad a temperature range as was reasonable was included in the background sampling. Specifically, we checked whether our background sampling scheme provided a temperature distribution at least as wide as that of the occurrences—a consideration important to ensure that the statistical model can ascertain thermal minima or maxima.

Due to the unequal historically unequal sampling effort for Insecta species in GBIF, the distribution of occurrence and background points was weighted towards North America and

Europe particularly for *Ae. aegypti* and *Ae. albopictus* (Figs E-F in S1 Text). If there are regional differences in the thermal minima and optima between continents, the estimates will be over-weighted towards North America and Europe. To address this potential bias, we additionally fit species distribution models for these two species dropping the occurrence and background points for North America and Europe and re-identifying thermal minima and optima to understand whether our estimates would change with different geographic representation in our dataset.

## Species distribution modeling

**XGBoost model fitting.**   Raster values for each occurrence and background centroid were extracted from the stack of covariates for a given species. The data was partitioned into a training and evaluation set, where 80% of the data was randomly selected for the training set and 20% was randomly selected for the validation set. We used stratified sampling so that there was an equal proportion of presence points in each set. We used gradient boosted classification trees to model the training data, as this class of algorithms fits flexible and nonlinear relationships (including thresholds) between environmental conditions and probability of species occurrence, identifies interactions between covariates due to the structure of the trees, and has been successfully used in other species distribution models [9,43,72]. Gradient boosted trees are a type of supervised learning algorithm that iteratively fits trees, each time fitting to the residual errors from the previous predictions, effectively ensembling weaker learners to accurately predict a target variable [73]. We fit models using eXtreme Gradient Boosting (XGBoost)[74] in R. Among the strengths of the XGBoost algorithm are its rapid computational speed when dealing with large-scale data, ability to handle class-imbalanced training sets (i.e., the number of observations are not equal across each level of the outcome variable), and ability to tolerate highly-collinear predictor variables. To quantify variation around model performance and output, we fit the model with 20 random train and validation splits.

**Bayesian hyperparameter optimization.**   Gradient boosted classification trees rely on hyperparameters that control how each tree is fit and how individual trees' estimates are combined. Bayesian optimization was conducted to tune key hyperparameters initialized with a range of user-inputted values, including learning rate (range: 0.0001 to 0.3), maximum tree depth (range: 2 to 50), minimum child weight (range: 1 to 50), subsample of observations used in each tree (range: 0.25 to 1), subsample of columns to use in each tree (range: 0.5 to 1), and minimum split loss (range: 0 to 100) [74,75]. For each round of the Bayesian optimization (up to a maximum of 24 rounds), we use 5-fold cross validation where we split the training data into 5 disjoint sets, and train the model on each combination of 4 folds while predicting out-of-sample for the fifth fold. We use the average out-of-sample log loss averaged over all 5 folds to identify the optimal number of boosting rounds (up to 10,000) for each iteration of the Bayesian optimization. After all iterations of the Bayesian optimization, we select the set of hyperparameters with the lowest out-of-sample log loss and train a final model using all 5 folds —representing all of the training data, which is 80% of the full data set.

**Interpreting the model.**   Receiver operating characteristic (ROC) curves and their area under the curve (AUC) values, which graphically and numerically illustrate the discrimination ability of a binary classifier, were computed for model predictions on both the training and evaluation sets. We were particularly interested in two aspects of the resulting models: the inferred relationship between the probability of occurrence and temperature, and the relative importance of temperature in predicting a species distribution, compared to other environmental covariates. To understand the relationship between temperature and occurrence, we plotted univariate partial dependence plots for temperature mean and temperature standard

deviation. Depending on the species, we use annual temperature, temperature during the photoperiod-dictated activity season, or temperature during the precipitation-dictated activity season. The partial dependence plots depict the marginal effect of temperature on the predicted value of the model and, at each temperature, is calculated as the mean predicted value (in this case, probability of species occurrence) across all combinations of other covariates observed in the dataset [76]. To assess the importance of temperature in relation to other environmental variables, measured in gain (i.e., the increase in accuracy that a given variable provides when a regression tree branch is split upon that variable) [75], we computed variable importance scores for the full set of predictors and compared them with temperature. We computed variable importance and PDPs for each bootstrapped iterations described above.

## Mosquito abundance curves from the lab

Physiological life history trait thermal performance curves were collected from published literature [16,29,31]. Briefly, in previously published work asymmetric responses like mosquito development rate (MDR) were modeled using a Briere function, and both concave-down symmetric responses like eggs per female per day (EFD) and probability of egg-to-adult survival (pEA) as well as concave-up symmetric responses like mortality rate (μ) were modeled using quadratic functions. 5,000 posterior draws were selected from each distribution, and mosquito abundance M(T) was calculated using Eq (1), which was originally derived in [6], adapted from [10]. Median and 95% credible intervals were computed for each species' M(T) and critical thermal limits.

$$M(T) = \frac{EFD(T)\ pEA\ (T)\ MDR(T)}{\mu(T)^2} \tag{1}$$

## Comparison of thermal relationships from XGBoost versus lab-based models of mosquito abundance

We compared thermal minima and optima of the partial dependence plots from the XGBoost model to the mosquito abundance M(T) curves from the lab trait trajectories. For species distribution models, thermal minima were identified as the temperature at which the partial dependence plot began increasing, which we operationalized as the first time the empirical derivative was positive and stayed positive for the next step in temperature as well (Fig 1C). Thermal minima could also correspond to the threshold temperature at which species probability of occurrence is increasing most rapidly, so we supplementarily identify thermal minima as the temperature where the empirical derivative is largest (Fig N in S1 Text). Thermal optima were identified as the point where the empirical derivative was zero and the partial dependence plot was at its maximum (Fig 1C). We did not identify thermal maxima, as few species had partial dependence plots that clearly declined after the thermal optima and then reached a lower plateau that was within the range of observed temperatures. For the mosquito abundance M(T) curves derived from laboratory studies, to determine the thermal minima, we identified the temperature at which M(T) exceeded zero for each M(T) curve calculated from posterior draws and then calculated the median, 2.5th and 97.5th percentiles to produce a central estimate and confidence interval for thermal minima. For thermal optima, we identified the temperature where M(T) was highest for each curve, and similarly calculated median and 95% confidence intervals. We calculate the Pearson correlation between the median species distribution model-based thermal minima and the median lab-based thermal minima, and repeat for thermal optima to quantify how well the two methods compare. In particular, because

temperature varies substantially in the field and M(T) curves are based on constant temperature measurements in the lab, we expect nonlinear averaging to affect the exact thermal limits observed in the field.

## Results

The species distribution models predicted the occurrence of all focal species with high accuracy: discrimination across both training (in-sample) and evaluation (out-of-sample) datasets produced an area under the receiver operating curve (AUC) above 0.9, where 1 indicates perfect discrimination of presence and absence and 0.5 is no better than a coin toss (Fig 2; Table C in S1 Text; Fig L in S1 Text). Of the focal species, *Cx. pipiens* recorded the lowest out-of-sample evaluation AUC of 0.935. The fact that the AUCs in the evaluation set were comparable to the training set suggests that the models are learning general patterns, differentiating between training and evaluation sets, and not overfitting to the training data.

Measures of temperature, i.e., either annual or activity season-limited (as appropriate) temperature mean and temperature standard deviation, were consistently important predictors of

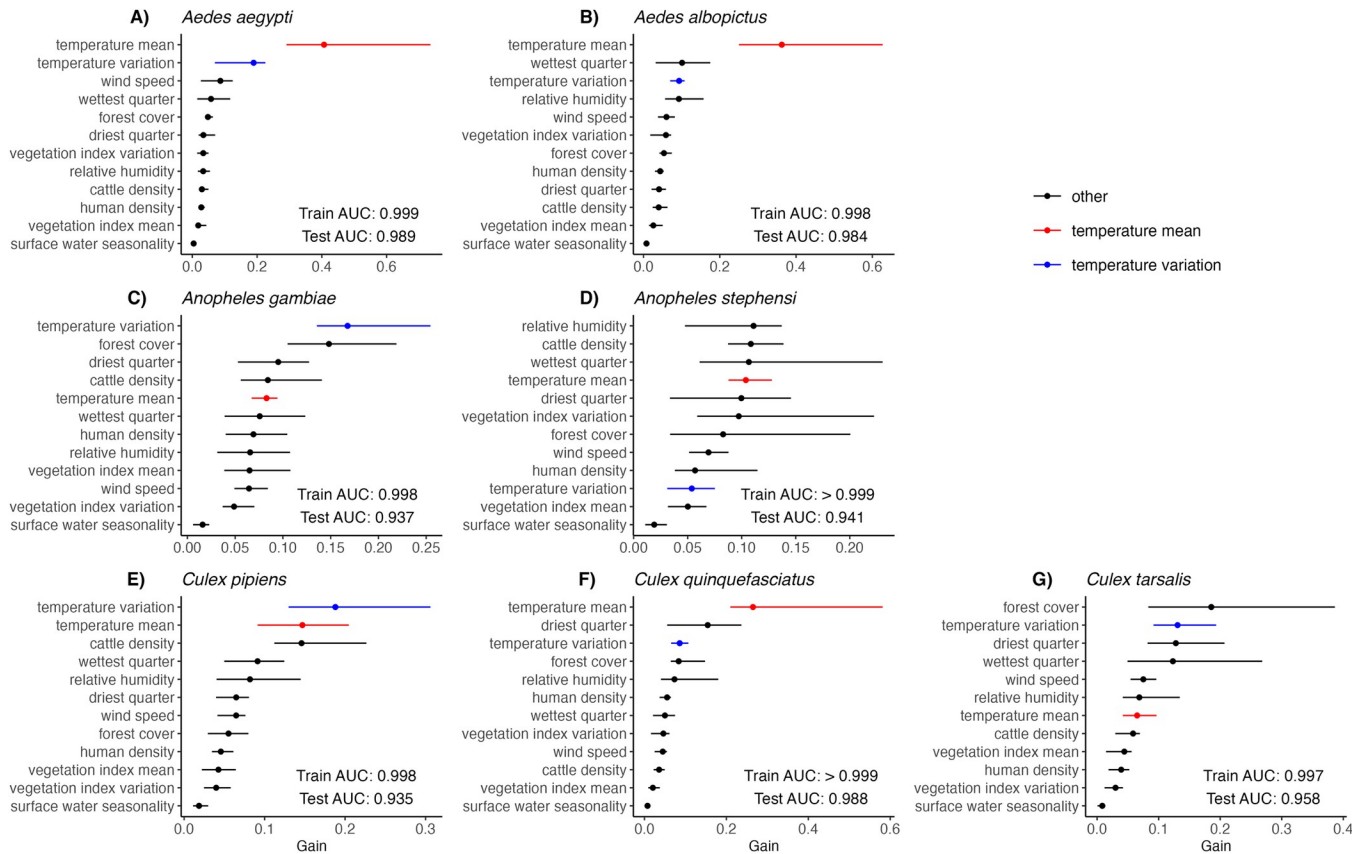

**Fig 2. Temperature mean (red) and standard deviation (blue) were important predictors of mosquito occurrence for all focal species.** Variable importance, measured in gain, is shown for each predictor variable by species. Temperature mean variables, including year-round temperature annual mean, photoperiod activity season temperature mean, and precipitation activity season temperature mean, are colored in red; only the most biologically appropriate temperature mean variable was included for each species (Table 1). Likewise, temperature standard deviation variables (Table 1) are colored in blue. Other predictors (black bars) include cattle density, enhanced vegetation index mean, enhanced vegetation index standard deviation, forest cover percentage, human population density, precipitation of the driest quarter, precipitation of the wettest quarter, relative humidity, surface water seasonality, and wind speed. Points represent the median importance across the 20 bootstrapped model iterations and lines represent the range over all model iterations. Test and training AUC values are similarly medians and full ranges over the model iterations.

mosquito occurrence, as measured by gain (Fig 2). For all species, mean temperature was among the top seven predictors. For *Ae. aegypti*, *Ae. albopictus*, and *Cx. quinquefasciatus* temperature mean was the top predictor and temperature standard deviation was second or third (Fig 2). *An. gambiae*, *Cx. pipiens*, and *Cx. tarsalis* had temperature standard deviation as a more important predictor than temperature mean (Fig 2).

Beyond temperature, different predictors were important for different mosquito species. Based on mean gain across the 20 bootstrapped iterations, forest cover was the most important predictor for *Cx. tarsalis* and second most important for *An. gambiae*, and relative humidity was most important for *An. stephensi* (Fig 2). Precipitation variables were among the top-5 predictor variables for all species. Precipitation of the wettest quarter was among the top-5 predictors for *Ae. aegypti*, *Ae. albopictus* and *Cx. pipiens* (Fig 2). Precipitation of the driest quarter was among the top-5 predictors for *An. gambiae* and *Cx. quinquefasciatus*, and both precipitation variables were among the top-5 predictors for *An. stephensi* and *Cx. tarsalis* (Fig 2). Surface water seasonality, however, was the least important predictor variable for all species distribution models (Fig 2).

While mechanistic relationships between temperature and mosquito abundance, based on laboratory-derived M(T), were unimodal and hump-shaped in form, the temperature relationships inferred from the statistical models (univariate XGBoost partial dependence plots; PDPs) were typically nonlinear with steep thresholds and plateaus (Fig 3). Both the mechanistic and statistical models showed steep increases as temperatures exceeded lower thermal limits (Fig 3). By contrast, while M(T) consistently responded unimodally to temperature, relationships from XGBoost PDPs mostly only showed lower thermal limits and thermal optima, but not upper limits (Fig 3). The difference in functional forms between M(T) and XGBoost PDPs is consistent with the fact that they each describe distinct processes. The mechanistic model, M(T), describes a relationship between temperature and mosquito *abundance* that continuously varies with temperature and the underlying laboratory-measured traits. By contrast, the PDPs from XGBoost capture the relationship between temperature and mosquito *occurrence probability*, which we expect to rise and fall sharply and reach a plateau at intermediate, suitable temperatures. The M(T) curves for all species showed clear thermal minima, optima, and maxima because the underlying laboratory experiments captured the full range of temperatures at which mosquito performance is optimized and at which it declines to zero.

Given that both the laboratory-based mechanistic approach (M(T)) and the statistical approach (XGBoost PDPs) predicted thermal minima and optima for each species, we asked whether the results of the two approaches were concordant. The estimates of thermal minima between M(T) and PDP were highly correlated across species (r = 0.869; Fig 4; Figs M-N in S1 Text), suggesting that M(T) captures key species-specific temperature constraints on occurrence in the field. Mosquito species of the *Aedes* genus exhibited thermal minima that were approximately 5°C cooler in the field than predicted in the laboratory (Fig 4; Fig M in S1 Text). *Cx. tarsalis* and *Cx. pipiens* consistently among the two lowest thermal minima, and *An. gambiae* and *An. stephensi* had the two highest thermal minimum when viewed across M(T) and PDP (Fig 4 and Figs M-N in S1 Text). The alternative identification of thermal minima as the threshold temperature with the greatest increase in species occurrence resulted in a similarly high correlation between M(T) and PDP-based thermal minima (r = 0.871), but resulted in thermal minima approximately 2–5°C warmer in the field than predicted by the laboratory for *Anopheles* mosquitoes (Fig N in S1 Text). Thermal optima were similar, although more weakly correlated, between M(T) and PDP estimates (r = 0.687), supporting, at least in part, the conclusion that field based observations capture key components of lab-based non-linear response to temperature. However, the relationship appeared to deviate the most for temperate species *Cx. pipiens* and *Cx. taralis*, for which thermal optima were 9.4°C and

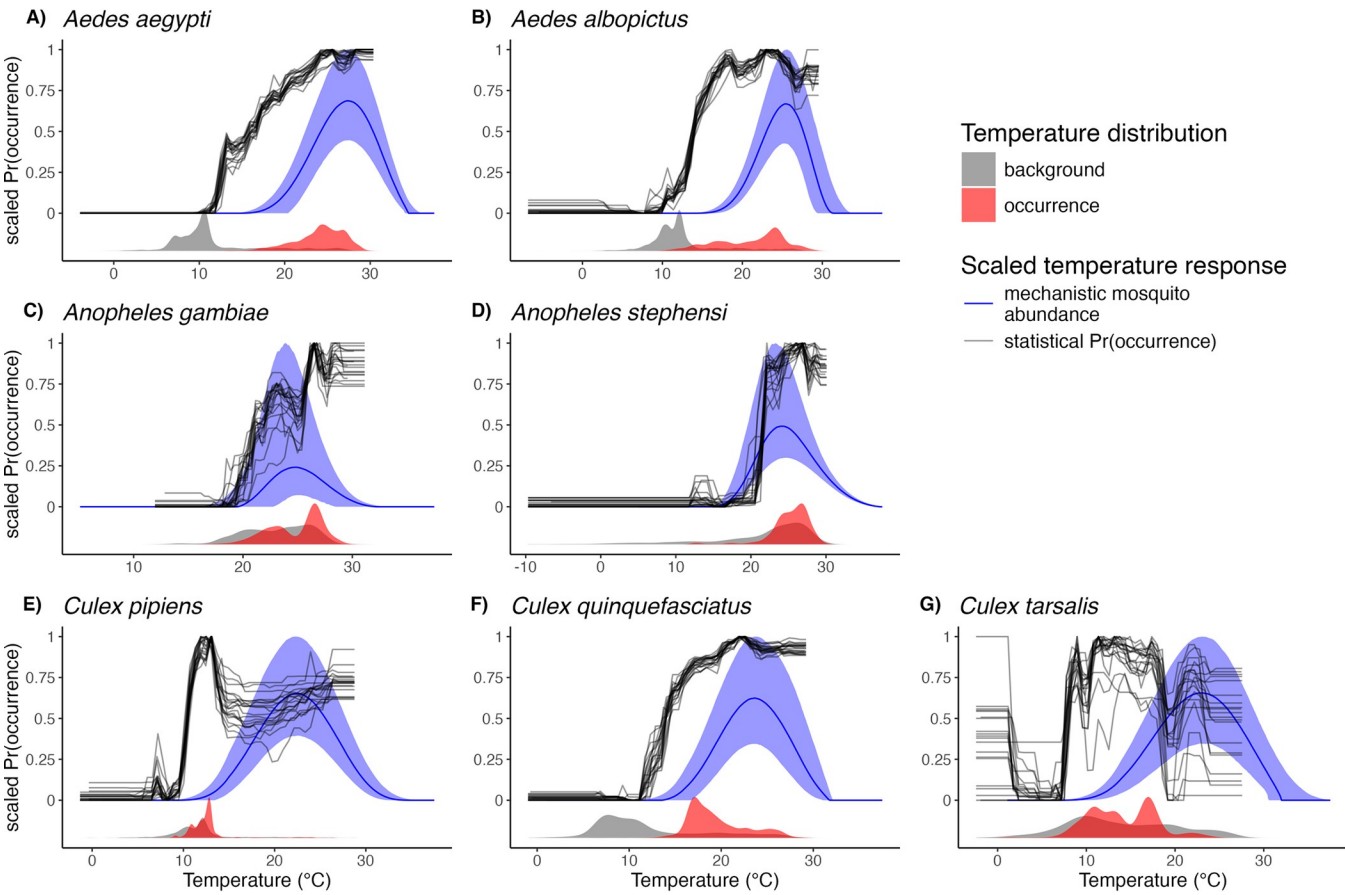

**Fig 3. Comparing mosquito temperature responses from mechanistic laboratory-based mosquito abundance models and XGBoost statistical models.**
Mosquito abundance for each species is estimated as the median M(T) curve with a shaded 95% credible interval band (blue lines and shaded ribbons). XGBoost covariate responses are univariate partial dependence plots (PDPs) for annual mean temperature, where mean temperature was bounded by photoperiod or precipitation for a subset of species, that show the marginal effects of temperature on model prediction (black lines). Each black line for the PDPs represents one of the 20 model iterations. Both M(T) and PDPSs are scaled to range from 0 to 1. Grey and red density plots in the bottom of each panel show the distribution of observed annual temperatures for background and occurrence points, respectively.

11.5˚C lower, respectively, in the field versus laboratory (Fig 4; Table D in S1 Text; Figs M-N in S1 Text). We could not compare thermal maxima because our XGBoost models did not identify thermal maxima for any focal species.

When fitting models for *Ae. aegypti* and *Ae. albopictus* without North America (Figs O-P in S1 Text) and Europe (Figs Q-R in S1 Text) occurrence and background points, we find little difference in detected thermal minima and optima or their correlations to the mechanistic lab-based estimates (Fig S in S1 Text), suggesting there are likely minimal differences in critical thermal values between geographic regions.

## Discussion

For all seven major vectors of human disease we investigated, species distribution models captured occurrence probability with high discrimination and accuracy and temperature was an important predictor (Fig 2; Fig L in S1 Text). In particular, temperature mean and standard deviation were highly important predictors across all focal species (Fig 2). This finding echoes the importance of temperature as a predictor in past models, although many such models define temperature differently from our study (e.g., temperature suitability, temperature of the

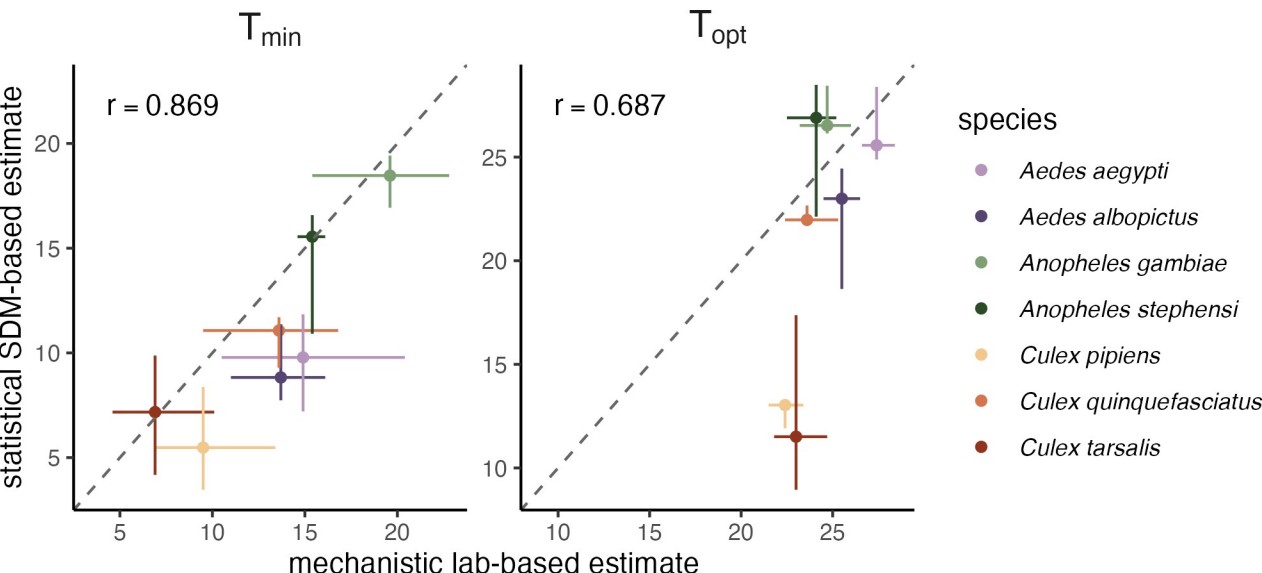

**Fig 4. Mechanistic models (M(T); x-axis) captured species' thermal minima observed in the field (XGBoost PDPs; y-axis).** The dashed diagonal line is the 1:1 line, and mosquito species are colored by genus. The Pearson correlation (r) was 0.869 for the PDP versus M(T) relationship for average $T_{min}$ and 0.687 for the PDP versus M(T) relationship for average $T_{opt}$. For mechanistic estimates of $T_{min}$ and $T_{opt}$, points indicate medians and horizontal line ranges are the 95% CI from the 5000 posterior samples used. For statistical estimates of $T_{min}$ and $T_{opt}$, points indicate medians and vertical lines are the full minimum to maximum range over the 20 bootstrapped model iterations.

hottest quarter, temperature of the coldest month) [43,48,77,78]. Four species (*Ae. aegypti*, *Ae. albopictus*, *Cx. pipiens*, and *Cx. quinquefasciatus*) had either temperature mean or standard deviation as their most important predictor (Fig 2). Nonlinear relationships with temperature from 'bottom-up' models of mosquito abundance from lab life-history traits were partially recapitulated in the 'top-down' statistical models of global-scale occurrence data, particularly at the lower end of the temperature range, consistent with previous work on other mosquito species [48]. In general, thermal minima and optima were highly comparable across species between the trait-based and statistical approaches (Fig 4). We find that the thermal minima are more strongly correlated than the thermal optima, consistent with the fact that the species distribution models aim to identify the conditions for species occurrence and the partial dependence plots show steep increases from low temperature not conducive to species occurrences then plateau. By contrast, PDPs show little change in occurrence probability with further changes in temperature. In contrast to the thermal minima and optima, thermal maxima were not comparable: XGBoost did not consistently detect thermal maxima for occurrence in the field, despite laboratory evidence that mosquito life history traits decline precipitously at high temperatures. This suggests that thermal constraints on species may not be identifiable using observational approaches, such as species distribution models, alone.

Thermal physiology theory and experiments have established that organismal performance is constrained to an operative range of temperatures at which key functions can occur and populations can stably persist [79–83]. Yet, realized species distributions may not clearly reflect these thermal constraints if other factors are also important for constraining distributions, including rainfall, seasonality, biotic interactions, habitat, movement rates, and the range of temperatures that occur within accessible regions [45,84–90]. For example, aridity may constrain mosquito distributions beyond direct high-temperature constraints. Likewise, average temperatures across activity seasons may not reach levels high enough to exclude mosquito persistence [7,79,91–93]. Alternatively, more extreme temperature variation at low or high

means can limit organismal performance, and even moderate variation can reduce performance near optimal temperatures (i.e., where M(T) curves are concave-down; Fig 3). The impact of temperature variation could explain the observation that for species with cooler thermal minima and optima, estimates at constant temperatures in the lab were up to 10°C warmer than estimates in the field (Fig 4).

This highlights the importance of climate change projections that consider both field-based estimates of thermal and other constraints on species distributions and more mechanistic estimates, particularly for thermal optima and maxima, which may be difficult to observe in the field under current and historical conditions [94]. Simply projecting species distribution models like those derived here under future climate change scenarios is likely to overlook the potential for warming temperatures to exceed thermal optima and limits for species persistence. Moreover, it is critical for studies of vector-borne disease transmission under future climate change to account for the gap between potential and realized future vector thermal niches, as vectors may not immediately track geographic shifts in climate suitability. Approaches that combine mechanistic thermal performance information (e.g., from laboratory studies and life history models) with statistical inference based on current distributions are most likely to accurately capture current and future environmental constraints on species persistence, including for noxious species like disease vectors.

This study incorporates several methodological innovations that advance upon previously published species distribution modeling methods for mosquitoes, particularly focused on estimating temperature bounds, with applications for climate change models. First, we computed covariates aligned to the time of collection of our mosquito occurrence data, in contrast to numerous studies that use climatologies estimated from the years preceding the occurrence point sampling time frame [9,43,47,49,95,96]. Second, given our explicit goal of estimating thermal limits, we restricted our measures of average temperature to the relevant activity season of each focal species [97], which ensures that we are using the temperature range of the mosquito in the field when that species is active and non-dormant (and therefore comparable to laboratory experimental measurements). Third, we selected background points from the ecoregions in which a species occurs plus a 200 km buffer, as well as adjacent ecoregions, to ensure that we are estimating temperature limitations based on regions that are plausible based on habitat suitability. This method, grounded in ecological theory [34,37,98] and similar to background limitation methods for non-mosquito species in the literature [41,99–102], functionally decouples habitat and temperature. This is largely a new advancement for mosquito species distribution modeling. Finally, while most mosquito species distribution modeling studies focus on identifying predictors of one or two mosquito species, here we explicitly aim to infer thermal limits and optima and to compare these to mechanistic estimates [77] among multiple medically-relevant mosquito vector species [43,78,103–105].

However, our study also has several limitations. First, the lack of thermal breadth in the background sampling may limit our ability to accurately infer thermal maxima. Mosquito species that occur in hotter climates, such as *Ae. aegypti*, *An. stephensi*, and *An. gambiae*, lack a sufficient amount of ecoregion-matched background points that have comparably high, or higher, temperatures than the occurrence points, making it difficult to estimate the temperature range that would be hot enough to prevent occurrence (i.e., identify thermal maxima) [106] and some species may not currently be constrained by their physiological upper thermal limits. Second, because our goal was to create accurate but comparable species distribution models for seven mosquito species across their global extent, we primarily relied on publicly available GBIF occurrence points (see Methods for data filtering criteria), which may not capture the full extent of each species that other sources such as local vector surveillance data might capture. Third, since we were focused on comparing laboratory-derived thermal

performance curves to field occurrence, we used temperatures measured during the activity season as predictors in XGBoost models for four out of seven species. Despite the importance of activity season temperature, other climatic variables such as winter low temperatures or dry-season aridity may be equally or more important limitations on species distributions. Fourth, the species distribution modeling framework assumes that species occupy their full climatic niche, but this may not be the case for species that are actively expanding their ranges. Finally, our analysis is based on a core assumption that using the 2000–2019 average of covariates can accurately characterize the average ecological or habitat preferences of a mosquito reported in a specific date and location.

## Conclusions

As climate change shifts the geographic and seasonal distribution of environmental conditions, it is critical to understand how temperature limits species ranges. Temperature in particular affects vector-borne diseases because of its effects on vector biting rate, parasite incubation rate, survival, and other life history traits. However, the thermal constraints on vector occurrence are less well understood, and particularly how they vary among important vector species. Here, we showed that temperature mean and variation during the activity season provide important constraints on the ranges of seven important mosquito vectors, and that thermal minima and, to a lesser degree, thermal optima observed in the field are closely correlated to those measured in laboratory experimental studies. This finding suggests that species distribution models can, to some extent, contribute to understanding the thermal biology of organisms that cannot be studied in a laboratory setting. Importantly, statistical species distribution models derived from field observations did not clearly identify upper thermal constraints even though these have been directly demonstrated experimentally in the laboratory. Climate change is likely to drive many regions past the currently observed range of temperatures. This highlights the critical need for mechanistic, trait-based studies that capture temperature ranges at which mosquito abundance and occurrence begin to decline [6,106,107] and, at the very least, emphasizes that extrapolations from statistical models based on current species distributions should be validated using physiological models [35,107,108].

## Supporting information

**S1 Text.  Table A: Filtered number of occurrences by species after each data cleaning step. Table B: Environmental covariates and respective data sources, accessed from Google Earth Engine. Table C: Model performance. Median and range of AUC (min-max) across bootstrapping iterations for both in-sample and out-of-sample performance. Table D: Thermal minima, optima, and maxima with rank ordering across species for the mechanistic trait based model (M(T)) versus the species distribution model (partial dependence plots (PDPs)). Fig A: Pairwise correlation plot of environmental predictors used in the model.** A pairwise correlation analysis was run for each set of two covariates, and any variables that had a correlation that exceeded the R < |0.8| threshold were reassessed and modified, or dropped from the analysis entirely. CD is cattle density; EVIM is enhanced vegetation index mean; EVISD is enhanced vegetation index standard deviation; FC is forest cover percentage; HPD is human population density; PDQ is precipitation of the driest quarter; PhotoASTM is photoperiod activity season temperature annual mean; PhotoASTSD is photoperiod activity season temperature standard deviation; PrecipASTM is precipitation activity season temperature annual mean; PrecipASTSD is precipitation activity season temperature standard deviation; PWQ is precipitation of the wettest quarter; SW is surface water seasonality; TAM is year-round temperature annual mean; TASD is year-round temperature annual standard

deviation; ARH is average relative humidity; and WS is wind speed. There are NAs among temperature variables as there was only one set included in each model (e.g., the *Ae. albopictus* model included PhotoASTM and PhotoASTD but not TAM, TASD, PrecipASTM, or PrecipASTD). **Fig B: Photoperiod activity season**. Map of the world shaded in with the length of the photoperiod activity season. **Fig C: Precipitation activity season.** Map of Africa shaded in with the length of the precipitation activity season in days. Source for precipitation data is described in S2 Table. **Fig D: Raster plots of all environmental covariates used in the model.** Geographic and spatial distribution of every covariate used in the model, including cattle density (CD), enhanced vegetation index mean (EVIM), enhanced vegetation index standard deviation (EVISD), forest cover percentage (FC), human population density (HPD), precipitation of the driest quarter (PDQ), photoperiod activity season temperature mean (PhotoASTM), photoperiod activity season temperature standard deviation (PhotoASTSD), precipitation activity season temperature mean (PrecipASTM), precipitation activity season temperature standard deviation (PrecipASTSD), precipitation of the wettest quarter (PWQ), surface water seasonality (SW), year-round temperature annual mean (TAM), year-round temperature annual standard deviation (TASD), average relative humidity (ARH), and wind speed (WS). PrecipASTM and PrecipASTSD are only depicted over Africa as this is the region the data was used (*An. gambiae*'s data points are restricted to Africa). **Fig E: Species occurrence and pseudo-absence background map for *Aedes aegypti*.** Species occurrence centroids (red) and associated pseudo-absence background centroids (black) are plotted, superimposed on the respective set of ecoregions in which their buffered occurrence centroids fall and adjacent ecoregions (gray). The background shapefiles are based on from https://ecoregions.appspot.com/ and coastlines are from https://ec.europa.eu/eurostat/web/gisco. **Fig F: Species occurrence and pseudo-absence background maps for *Aedes albopictus*.** Species occurrence centroids (red) and associated pseudo-absence background centroids (black) are plotted, superimposed on the respective set of ecoregions in which their buffered occurrence centroids fall and adjacent ecoregions (gray). The background shapefiles are based on from https://ecoregions.appspot.com/ and coastlines are from https://ec.europa.eu/eurostat/web/gisco. **Fig G: Species occurrence and pseudo-absence background maps for *Anopheles stephensi*.** Species occurrence centroids (red) and associated pseudo-absence background centroids (black) are plotted, superimposed on the respective set of ecoregions in which their buffered occurrence centroids fall and adjacent ecoregions (gray). The background shapefiles are based on from https://ecoregions.appspot.com/ and coastlines are from https://ec.europa.eu/eurostat/web/gisco. **Fig H: Species occurrence and pseudo-absence background maps for *Anopheles gambiae*.** Species occurrence centroids (red) and associated pseudo-absence background centroids (black) are plotted, superimposed on the respective set of ecoregions in which their buffered occurrence centroids fall and adjacent ecoregions (gray). The background shapefiles are based on from https://ecoregions.appspot.com/ and coastlines are from https://ec.europa.eu/eurostat/web/gisco. **Fig I: Species occurrence and pseudo-absence background maps for *Culex tarsalis*.** Species occurrence centroids (red) and associated pseudo-absence background centroids (black) are plotted, superimposed on the respective set of ecoregions in which their buffered occurrence centroids fall and adjacent ecoregions (gray). The background shapefiles are based on from https://ecoregions.appspot.com/ and coastlines are from https://ec.europa.eu/eurostat/web/gisco. **Fig J: Species occurrence and pseudo-absence background maps for *Culex quinquefasciatus*.** Species occurrence centroids (red) and associated pseudo-absence background centroids (black) are plotted, superimposed on the respective set of ecoregions in which their buffered occurrence centroids fall and adjacent ecoregions (gray). The background shapefiles are based on from https://ecoregions.appspot.com/ and coastlines are from https://ec.europa.eu/eurostat/web/gisco. **Fig K: Species occurrence and pseudo-absence background**

**maps for *Culex pipiens*.** Species occurrence centroids (red) and associated pseudo-absence background centroids (black) are plotted, superimposed on the respective set of ecoregions in which their buffered occurrence centroids fall and adjacent ecoregions (gray). The background shapefiles are based on from https://ecoregions.appspot.com/ and coastlines are from https://ec.europa.eu/eurostat/web/gisco. **Fig L: XGBoost models accurately predicted mosquito occurrence in- and out-of-sample.** Receiver operating characteristic (ROC) curves and area under the curve (AUC) values for assessment of model discrimination are depicted, where 1 represents perfect discrimination of presence and absence and 0.5 represents discrimination no better than chance. Curves depict the training set (red) and the evaluation set (blue). **Fig M: Thermal minima and thermal optima derived from the PDPs.** Thermal minima were identified as the temperature at which the partial dependence plot began increasing, which we operationalized as the first time the empirical derivative was positive and stayed positive for the next step in temperature as well (Fig 1C). Thermal optima were identified as the point where the empirical derivative was zero and the partial dependence plot was at its maximum (Fig 1C). We did not identify thermal maxima, as few species had partial dependence plots that clearly declined after the thermal optima and then reached a lower plateau that was within the range of observed temperatures. Central thermal minima and optima values are medians across the 20 model iterations, and parentheses indicate the full range of values over the model iterations. **Fig N: Comparison between statistical and mechanistic under alternative definition of $T_{min}$.** Lower thermal limits are instead identified as the temperature with the largest empirical derivative in the PDP. Panel a is as in Fig M. Panel b is as in Fig 4. **Fig O: Species occurrence and pseudo-absence background map for *Aedes aegypti* without North America.** Species occurrence centroids (red) and associated pseudo-absence background centroids (black) are plotted, superimposed on the respective set of ecoregions in which their buffered occurrence centroids fall and adjacent ecoregions (gray). The background shapefiles are based on from https://ecoregions.appspot.com/ and coastlines are from https://ec.europa.eu/eurostat/web/gisco. **Fig P: Species occurrence and pseudo-absence background map for *Aedes albopictus* without North America.** Species occurrence centroids (red) and associated pseudo-absence background centroids (black) are plotted, superimposed on the respective set of ecoregions in which their buffered occurrence centroids fall and adjacent ecoregions (gray). The background shapefiles are based on from https://ecoregions.appspot.com/ and coastlines are from https://ec.europa.eu/eurostat/web/gisco. **Fig Q: Species occurrence and pseudo-absence background map for *Aedes aegypti* without Europe.** Species occurrence centroids (red) and associated pseudo-absence background centroids (black) are plotted, superimposed on the respective set of ecoregions in which their buffered occurrence centroids fall and adjacent ecoregions (gray). The background shapefiles are based on from https://ecoregions.appspot.com/ and coastlines are from https://ec.europa.eu/eurostat/web/gisco. **Fig R: Species occurrence and pseudo-absence background map for *Aedes albopictus* without Europe.** Species occurrence centroids (red) and associated pseudo-absence background centroids (black) are plotted, superimposed on the respective set of ecoregions in which their buffered occurrence centroids fall and adjacent ecoregions (gray). The background shapefiles are based on from https://ecoregions.appspot.com/ and coastlines are from https://ec.europa.eu/eurostat/web/gisco. **Fig Q: Critical thermal minima and optima identified with different geographic samples.** a) Scaled probability of occurrence (top row) and the derivative of probability of occurrence (bottom row) from partial dependence plots for the full sample (grey lines and circles), without Europe (red lines and squares) and without North America (blue lines and triangles). Text labels between panels indicates the median, minimum, and maximum for the thermal minima ($T_{min}$) and thermal optima ($T_{opt}$) across the 20 model iterations. b) As in Fig 4, but showing $T_{min}$ (left panel) and $T_{opt}$ (right panel) from the three different geographic samples (all occurrences

[circles], without Europe [squares], and without North America [triangles]). Each panel displays the Pearson's correlation for the different geographic samples.
(PDF)

## Acknowledgments

The authors thank Marta Shocket and Oswaldo Villena for providing mosquito life history trait data, and Marianne Sinka and Joshua Longbottom for supplying additional species occurrence points for *Anopheles stephensi*.

## Author Contributions

**Conceptualization:** Tejas S. Athni, Marissa L. Childs, Erin A. Mordecai.

**Data curation:** Tejas S. Athni, Marissa L. Childs.

**Formal analysis:** Tejas S. Athni, Marissa L. Childs, Caroline K. Glidden.

**Funding acquisition:** Tejas S. Athni, Erin A. Mordecai.

**Investigation:** Tejas S. Athni.

**Methodology:** Tejas S. Athni, Marissa L. Childs, Caroline K. Glidden.

**Project administration:** Erin A. Mordecai.

**Supervision:** Marissa L. Childs, Erin A. Mordecai.

**Validation:** Tejas S. Athni, Marissa L. Childs, Caroline K. Glidden.

**Visualization:** Marissa L. Childs.

**Writing – original draft:** Tejas S. Athni, Marissa L. Childs.

**Writing – review & editing:** Tejas S. Athni, Marissa L. Childs, Caroline K. Glidden, Erin A. Mordecai.

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
