## [Decision Letter · Decision Letter 0]

27 Mar 2024

Dear Dr. Glidden,

Thank you very much for submitting your manuscript "Temperature dependence of mosquitoes: comparing mechanistic and machine learning approaches" for consideration at PLOS Neglected Tropical Diseases. As with all papers reviewed by the journal, your manuscript was reviewed by members of the editorial board and by several independent reviewers. In light of the reviews (below this email), we would like to invite the resubmission of a significantly-revised version that takes into account the reviewers' comments. 

Most of the reviewers have expressed major concerns and reservations on fundamental aspects of experimental design and approaches that have significant impact on your findings. These concerns must be fully addressed for your MS to be considered for publication.

We cannot make any decision about publication until we have seen the revised manuscript and your response to the reviewers' comments. Your revised manuscript is also likely to be sent to reviewers for further evaluation.

Sincerely,

Paul O. Mireji, PhD

Section Editor

Paul Mireji

Section Editor

Most of the reviewers have expressed major concerns and reservations on fundamental aspects of experimental design and approaches that have significant impact on your findings. These concerns must be fully addressed for your MS to be considered for publication.

Reviewer's Responses to Questions

**Key Review Criteria Required for Acceptance?**

**Methods**

-Are the objectives of the study clearly articulated with a clear testable hypothesis stated?

-Is the study design appropriate to address the stated objectives?

-Is the population clearly described and appropriate for the hypothesis being tested?

-Is the sample size sufficient to ensure adequate power to address the hypothesis being tested?

-Were correct statistical analysis used to support conclusions?

-Are there concerns about ethical or regulatory requirements being met?

Reviewer #1: Review comments.

Manuscript PNTD-23-01598 tiled: “Temperature dependence of mosquitoes: comparing mechanistic and machine learning. 

This study discusses the increasing global public health concern posed by mosquito vectors (e.g., Aedes, Anopheles, Culex spp.), which transmit diseases like dengue, Zika, chikungunya, West Nile, and malaria. The authors argues that mosquitoes are shifting geographically due to climate change and other human activities. As ectotherms, mosquitoes are highly sensitive to temperature, affecting their life history traits (like biting rate and survival probability), which show upper and lower thermal limits and intermediate optima in lab studies. According to the authors, the correlation between lab-based thermal responses and mosquitoes' responses in natural settings is unclear. To bridge this knowledge gap, the study used machine learning models based on thousands of global mosquito occurrences and high-resolution satellite data to estimate vector thermal responses. This approach, which included adjustments for mosquito activity season and ecologically relevant spatial sampling, revealed a strong correlation between laboratory-estimated thermal minima and field observations (r = 0.90), with a moderate correlation for thermal optima (r = 0.69). However, thermal maxima were not detectable in field distributions for comparison with lab estimates. The study concluded that lab studies can effectively predict lower thermal limits and optima of mosquitoes in the field. Additionally, lab-based models might capture physiological limits at high temperatures, crucial for understanding mosquito responses to climate change, which are not apparent in field observations.

First impression: The study title “Temperature dependence of mosquitoes: comparing mechanistic and machine learning. I am contemplating whether it is appropriate to draw comparisons between methodologies that fundamentally differ from each other. For instance, mechanistic models are process-driven and are typically calibrated using data derived from controlled biological experiments. These models have a clear and traceable logic in how they process information, closely following biological phenomena as observed in laboratory settings. In contrast, machine learning models often function as 'black boxes.' Their internal workings in processing data are not transparent, making it challenging to understand precisely how they arrive at their outputs. Furthermore, these models, primarily developed from extensive datasets, may lack a direct linkage to biological or ecological principles. They are designed to identify patterns and make predictions based on the data they are fed, without necessarily incorporating the underlying biological or ecological mechanisms. Therefore, comparing these two types of models might overlook the inherent differences in their approaches, purposes, and the nature of the data they are based on. While each has its strengths, they operate on different premises – mechanistic models with a focus on process and understanding, and machine learning models with an emphasis on pattern recognition and prediction.

Critical comments on the methodology used.

While the study delves into a compelling and potentially significant area of research, I have reservations regarding its methodology. In the case of mosquito studies, laboratory experiments are conducted to replicate the biophysical mechanisms defining and characterizing mosquito species. These experiments aim to understand species' responses to environmental factors like temperature, focusing on the process rather than being purely data-driven. The models or equations developed to estimate the lower and upper limits of a species' developmental stages are process-bound and characteristic of each species, adhering to principles that define them. However, translating these laboratory findings to natural settings using machine learning (ML) models presents significant challenges. Discrepancies between ML predictions and laboratory findings, particularly at lower thermal optima and upper limits, can arise from various factors. The complexity of natural environments, with factors like microclimates and ecological interactions, may not be fully captured in lab settings. ML models, despite their power, depend on the quality and range of input data, which might not comprehensively represent natural conditions. Laboratory studies often simplify complex biophysical mechanisms for practicality, potentially leading to gaps when applying these rules to real-world scenarios. Generalizing lab findings to field conditions via ML might fail to account for the nuanced dynamics of mosquito ecology. Therefore, while ML holds promise in bridging lab and field studies, its application needs careful consideration and calibration, respecting the complexities of ecosystems and the inherent limitations of lab experiments and ML algorithms.

Translating laboratory findings to natural settings using machine learning (ML) models can be challenging. This is exemplified by the varying correlation levels between ML predictions and laboratory findings, particularly regarding lower thermal optima and upper limits (thermal minima and field observations showing a high correlation of r = 0.90, but a more moderate correlation for thermal optima at r = 0.69). Several factors contribute to this discrepancy:

i. Complexity of natural environments: Mosquito habitats in nature are far more complex and varied than those in laboratory settings. Factors such as microclimates, ecological interactions, and geographical diversity significantly influence mosquito behavior and survival. These elements are often not fully replicated or captured in controlled laboratory environments.

ii. Limitations of machine learning models: ML models are highly dependent on the quality and range of input data. If laboratory data do not encompass the full spectrum of natural conditions or omit essential environmental variables, these models may fall short in accurately predicting real-world scenarios.

iii. Biophysical Mechanism Simplification: For practicality, laboratory studies often simplify the complex biophysical mechanisms of mosquito species. This necessary simplification for in-depth study can create gaps when ML models attempt to apply these rules to the more intricate conditions of the real world.

iv. Generalization from Laboratory to Field: While laboratory studies are crucial for grasping the basic biology of mosquitoes, extending these findings to field conditions via ML can potentially miss the subtle and dynamic aspects of mosquito ecology in nature.

Although ML offers a valuable means to connect laboratory research and field observations, its application should be thoughtfully considered. This involves recognizing the intricacies of natural ecosystems and the inherent limitations of both laboratory methodologies and ML algorithms.

Reviewer #2: The study elucidates to develop a model based on newer tools, on the occurrence of 7 important vector species belonging to Culicidae, in relation to temperature globally. However, the species distribution data used for the modelling study relies on GBIF, which remains still as a baseline data to describe the occurrence of these species globally. The authors could have chosen curated different Country wise data available in the literature. Even though the authors aim the study to be a global one, they mostly restrict the data to American (and to a lesser extent African) Countries. Vector borne diseases are mainly a problem of tropical countries, which remains the worst affected. Species occurrence data used in the study (Fig. 1a) in highly affected country by the disease, where the species they concentrate is mostly Asian and African Countries, where data on the occurrence of these species is shown as meagre.

In addition, the parameter chosen for this modelling is only a single environmental parameter, temperature. Even though it remains crucial in mosquito survival and development, other very important parameter relative humidity which is also a very crucial one is somehow not included at all. Earlier authors, used to compute a parameter saturation deficit ( a combination of both temperature and relative humidity) as both these parameters have been determined as the most important environmental parameters affecting mosquito survival and thereby their distribution.

In page 5 authors emphasis on the selection of the period of distribution as "active season". Active season is there only in temperate regions, in in tropical countries. Hope this is a global investigation. - May be modified

Reviewer #3: Methods are generally valid but need additional information. In particular, I would suggest the authors to provide more details regarding the XGBoost. One thing that is not clear to me is that how the XGboost infers the relationship between the probability of occurrence and temperature. It may be helpful if the authors can provide sort of schematic plots to help readers who are less familiar with that particular method.

**Results**

-Does the analysis presented match the analysis plan?

-Are the results clearly and completely presented?

-Are the figures (Tables, Images) of sufficient quality for clarity?

Reviewer #1: The confusion in this study arises from the observed discrepancy in accurately predicting thermal minima and thermal optima for the same mosquito species. It is perplexing how the machine learning models can predict one variable (thermal minima) with high accuracy (r = 0.90) but show less precision (r = 0.69) in predicting the other (thermal optima), despite both being characteristics of the same species. From a biophysical standpoint, this inconsistency seems counterintuitive since both limits are integral traits of the species, influenced by similar biological processes. The fact that these thermal characteristics, both resulting from and driven by the same biological processes, show different levels of predictability challenges the logical coherence of the study's findings. This inconsistency raises questions about the underlying methodologies or data used in the study, suggesting a need for a more nuanced approach that considers the interconnected nature of these biophysical traits.

Reviewer #2: Analysis had been carried out for the data they have chosen for the study. However, this is not comprehensive.

Reviewer #3: Results are clearly presented.

**Conclusions**

-Are the conclusions supported by the data presented?

-Are the limitations of analysis clearly described?

-Do the authors discuss how these data can be helpful to advance our understanding of the topic under study?

-Is public health relevance addressed?

Reviewer #1: Another challenge in this study also lies in the selection of covariates for building the machine learning (ML) model. The criteria or process used to choose variable 'x' is not clearly articulated, raising concerns about the foundation upon which the ML model was developed. Before delving into the complexities of an ML model, which often functions as a 'black box', it is essential to engage in what I refer to as "data exploration." This process involves a thorough examination of each variable, particularly environmental ones, to understand their individual and collective contributions to the phenomena we aim to predict. Data exploration is crucial as it helps in identifying the most relevant predictors and understanding the underlying relationships within the data. This preliminary step is vital for ensuring that the ML model is built on a solid and transparent foundation, enhancing its predictive accuracy and reliability. Without this initial exploration, there's a risk of overlooking key variables or misinterpreting their importance, which could lead to less effective models and questionable conclusions

Reviewer #2: Authors made an investigation to model the influence of temperature on the occurrence and distribution of 7 important species of mosquitoes. Their conclusion seems to be valid. They could arrive at therma minima and therma optima values. However they could not obtain a significant correlation for therma maxima. This could be owing to lack of including another equally important parameter, relative humidity for generating their model.

Reviewer #3: Conclusions are well justified.

**Editorial and Data Presentation Modifications?**

Reviewer #1: Reject

Reviewer #2: There are some errors such as

1) When a species is mentioned in the manuscript for the first instance those should be written in full and not as abbreviation. Page 4 lines 100-101

2) Culex quinquefasciatius, one of the species they include in the study is the main vector for Lymphatic filariasis in tropical Countries. Authors mention it as a arbo-viral vector only. Page 4 Lines 106-107.

Reviewer #3: Minor Revision

**Summary and General Comments**

Reviewer #1: This study discusses the increasing global public health concern posed by mosquito vectors (e.g., Aedes, Anopheles, Culex spp.), which transmit diseases like dengue, Zika, chikungunya, West Nile, and malaria. The authors argues that mosquitoes are shifting geographically due to climate change and other human activities. As ectotherms, mosquitoes are highly sensitive to temperature, affecting their life history traits (like biting rate and survival probability), which show upper and lower thermal limits and intermediate optima in lab studies. According to the authors, the correlation between lab-based thermal responses and mosquitoes' responses in natural settings is unclear. To bridge this knowledge gap, the study used machine learning models based on thousands of global mosquito occurrences and high-resolution satellite data to estimate vector thermal responses. This approach, which included adjustments for mosquito activity season and ecologically relevant spatial sampling, revealed a strong correlation between laboratory-estimated thermal minima and field observations (r = 0.90), with a moderate correlation for thermal optima (r = 0.69). However, thermal maxima were not detectable in field distributions for comparison with lab estimates. The study concluded that lab studies can effectively predict lower thermal limits and optima of mosquitoes in the field. Additionally, lab-based models might capture physiological limits at high temperatures, crucial for understanding mosquito responses to climate change, which are not apparent in field observations.First impression: The study title “Temperature dependence of mosquitoes: comparing mechanistic and machine learning. I am contemplating whether it is appropriate to draw comparisons between methodologies that fundamentally differ from each other. For instance, mechanistic models are process-driven and are typically calibrated using data derived from controlled biological experiments. These models have a clear and traceable logic in how they process information, closely following biological phenomena as observed in laboratory settings. In contrast, machine learning models often function as 'black boxes.' Their internal workings in processing data are not transparent, making it challenging to understand precisely how they arrive at their outputs. Furthermore, these models, primarily developed from extensive datasets, may lack a direct linkage to biological or ecological principles. They are designed to identify patterns and make predictions based on the data they are fed, without necessarily incorporating the underlying biological or ecological mechanisms. Therefore, comparing these two types of models might overlook the inherent differences in their approaches, purposes, and the nature of the data they are based on. While each has its strengths, they operate on different premises – mechanistic models with a focus on process and understanding, and machine learning models with an emphasis on pattern recognition and prediction.

Reviewer #2: In summary, if the authors would have used a curated data on the occurrence and distribution of the concerned species as well as if they would have included relative humidity, in addition to temperature, into the environmental parameters, they could have come out with a more reliable model on the influence of climatic and environmental parameters on the distribution of these species.

Reviewer #3: This is a clearly written paper but additional info on the method would be helpful.

PLOS authors have the option to publish the peer review history of their article (what does this mean?). If published, this will include your full peer review and any attached files.

Reviewer #1: No

Reviewer #2: Yes: DR N PRADEEP KUMAR

Reviewer #3: No
---

## [Decision Letter · Decision Letter 1]

27 Aug 2024

Dear Dr. Glidden,

We are pleased to inform you that your manuscript 'Temperature dependence of mosquitoes: comparing mechanistic and machine learning approaches' has been provisionally accepted for publication in PLOS Neglected Tropical Diseases.

Best regards,

Paul O. Mireji, PhD

Section Editor

Paul Mireji

Section Editor

Reviewer's Responses to Questions

**Key Review Criteria Required for Acceptance?**

**Methods**

-Are the objectives of the study clearly articulated with a clear testable hypothesis stated?

-Is the study design appropriate to address the stated objectives?

-Is the population clearly described and appropriate for the hypothesis being tested?

-Is the sample size sufficient to ensure adequate power to address the hypothesis being tested?

-Were correct statistical analysis used to support conclusions?

-Are there concerns about ethical or regulatory requirements being met?

Reviewer #2: Article has been modified according to the comments of this reviewer, and is hence acceptable for publication

Reviewer #3: (No Response)

**Results**

-Does the analysis presented match the analysis plan?

-Are the results clearly and completely presented?

-Are the figures (Tables, Images) of sufficient quality for clarity?

Reviewer #2: Article has been modified according to the comments of this reviewer, and is hence acceptable for publication

Reviewer #3: (No Response)

**Conclusions**

-Are the conclusions supported by the data presented?

-Are the limitations of analysis clearly described?

-Do the authors discuss how these data can be helpful to advance our understanding of the topic under study?

-Is public health relevance addressed?

Reviewer #2: Article has been modified according to the comments of this reviewer, and is hence acceptable for publication

Reviewer #3: (No Response)

**Editorial and Data Presentation Modifications?**

Reviewer #2: Please modify line 299 "An. Gambiae" to "An. gambiae"

Reviewer #3: (No Response)

**Summary and General Comments**

Reviewer #2: Article has been modified according to the comments of this reviewer, and is hence acceptable for publication

Reviewer #3: My previous comments are addressed.

PLOS authors have the option to publish the peer review history of their article (what does this mean?). If published, this will include your full peer review and any attached files.

Reviewer #2: **Yes: **N PRADEEP KUMAR

Reviewer #3: No

---

## [Editor Report · Acceptance letter]

7 Sep 2024

Dear Dr. Glidden,

We are delighted to inform you that your manuscript, "Temperature dependence of mosquitoes: comparing mechanistic and machine learning approaches," has been formally accepted for publication in PLOS Neglected Tropical Diseases.

Best regards,

Shaden Kamhawi

co-Editor-in-Chief

Paul Brindley

co-Editor-in-Chief
